# Application of an Enzymatic Hydrolysed L-α-Amino Acid Based Biostimulant to Improve Sunflower Tolerance to Imazamox

**DOI:** 10.3390/plants11202761

**Published:** 2022-10-19

**Authors:** Eloy Navarro-León, Elisabet Borda, Cándido Marín, Nuria Sierras, Begoña Blasco, Juan M. Ruiz

**Affiliations:** 1Department of Plant Physiology, Faculty of Sciences, University of Granada, 18071 Granada, Spain; 2R & D Plant Health, Bioiberica S.A.U., 08389 Barcelona, Spain

**Keywords:** amino acids, biostimulant, glutathione, herbicide, safener, sunflower

## Abstract

Herbicides, commonly used in agriculture to control weeds, often cause negative effects on crops. Safeners are applied to reduce the damage to crops without affecting the effectiveness of herbicides against weeds. Plant biostimulants have the potential to increase tolerance to a series of abiotic stresses, but very limited information exists about their effects on herbicide-stressed plants. This study aims to verify whether the application of a potential safener such as Terra-Sorb^®^, an L-α-amino acid-based biostimulant, reduces the phytotoxicity of an Imazamox-based herbicide and to elucidate which tolerance mechanisms are induced. Sunflower plants were treated with Pulsar^®^ 40 (4% Imazamox) both alone and in combination with Terra-Sorb^®^. Plants treated with the herbicide in combination with Terra-Sorb^®^ showed higher growth, increased acetolactate synthase (ALS) activity, and amino acid concentration with respect to the plants treated with Imazamox alone. Moreover, the biostimulant protected photosynthetic activity and reduced oxidative stress. This protective effect could be due to the glutathione S-transferase (GST) induction and antioxidant systems dependent on glutathione (GSH). However, no effect of the biostimulant application was observed regarding phenolic compound phenylalanine ammonium-lyase (PAL) activity. Therefore, this study opens the perspective of using Terra-Sorb^®^ in protecting sunflower plants against an imazamox-based herbicide effect.

## 1. Introduction

Weeds are included among the main biotic limitations that affect optimal crop productivity. Weeds establish a competitive relationship with the main crop to obtain water and nutrients essential for the growth and development of plants. This competition may reduce the yield and quality of the crops by more than 50% [1,2]. Herbicide application is currently the most widely used and efficient agricultural practice to restrict the proliferation of weeds. Herbicides are biologically active organic compounds that alter the basic physiological processes of plants, causing their death. For instance, the mechanism of action of the Imazamox [(5-(methoxymethyl)-2-(4-methyl-5-oxo-4-propan-2-yl-1H-imidazol-2-yl) pyridine-3-carboxylic acid)] herbicide is the inhibition of acetolactate synthase (ALS) enzymes. ALS is necessary for the synthesis of essential branched-chain amino acids, such as valine, leucine, and isoleucine, necessary for plant growth [3,4].

Herbicides must meet two antagonistic criteria: on the one hand, they must allow for the efficient control of a wide range of weed species and, on the other hand, be harmless or less phytotoxic for the crops they are applied to [5]. The most characteristic visible symptoms of phytotoxicity that appear a few days after the application of herbicides on plants are an accumulation of anthocyanins, chlorosis, and necrotic spots on the leaves [6]. Different studies have shown that herbicides restrict plant growth and development through increased oxidative stress and decreased antioxidant and photosynthetic activities. Thus, a wide range of herbicide types causes oxidative stress-inducing the accumulation of reactive oxygen species (ROS) such as H_2_O_2_ and O_2_^−^ [7,8]. This phenomenon was observed in plants treated with ALS-inhibiting herbicides such as mesosulfuron- methyl, although there are a lack of studies about the effect of imazamox herbicides on oxidative stress [9].

Concerning photosynthesis, the analysis of chlorophyll (Chl) *a* fluorescence is a powerful tool for studying this process and the effects of stress on it [10]. The photosynthetic process is initiated when light is absorbed by the pigments of the membrane antenna complexes. Part of the absorbed energy is transferred as excitation energy and trapped by the reaction centers (RCs), where it is useful for photosynthetic reactions, and the other part is dissipated as heat and low-energy light (fluorescence) [11]. In Chl *a* fluorescence analysis, a dark-adapted leaf with the photosystem II (PSII) electron-acceptor pools completely oxidized was illuminated with a high-intensity light source. This illumination produced a rapid polyphasic rise of fluorescence until the electron acceptors in the photosystems were reduced. This polyphasic curve of fluorescence presented several phases (O, J, I, P). Derived from this curve, multiple parameters indicative of the photosystem functioning were available [10]. Several studies showed that the analysis of Chl *a* fluorescence is very useful for estimating the damage caused by Imazamox-based herbicides, although they do not directly affect the photosynthesis process [12,13].

Certain compounds called protectors or “safeners” are usually applied together with herbicides to reduce phytotoxicity. Safeners are applied to reduce damage to crops without affecting the effectiveness of herbicides against weeds [5]. Currently, the use of herbicides associated with “safeners” represents roughly 30% of the value of herbicide sales [14]. Safeners induce the resistance responses of plants to herbicide toxicity through the following mechanisms: retention on the leaf surface to avoid herbicide drift, decreased absorption and translocation, desensitization of the herbicide target protein, and detoxification or herbicide metabolism in plants [6]. This latter mechanism of herbicide resistance involves the performance of a series of physiological processes called non-target site resistance (NTSR). Two important plant mechanisms of the NTSR are glutathione (GSH) homeostasis and phenolic compounds [15].

GSH is a tripeptide made up of three amino acids: glutamic acid, cysteine, and glycine. The synthesis and availability of these amino acids are essential for reducing the toxic effects of herbicides. Thus, GSH, together with the action of the enzyme glutathione S-transferase (GST), produces the herbicide conjugation with GSH, forming a complex that later facilitates the degradation of the herbicide into non-toxic compounds for plants [16]. Furthermore, GSH acts directly in the detoxification of toxic radicals produced by herbicides in plants, such as H_2_O_2_ and methylglyoxal, through the induction of GSH-peroxidase enzymes and the glyoxalase system [17,18].

In addition, the application of phenolics in safeners produces an improvement in sensitive plants to the resistance of herbicides [19,20]. Salicylic acid is a phenolic compound that acts as a hormone and induces the antioxidant response of plants, which could explain its protective effect. Furthermore, the concentration of phenolic compounds increases very significantly in plants resistant to the Imazamox herbicide, enhancing the plant antioxidant response. Hence, the direct use of salicylic acid and other polyphenols as safeners, or the use of compounds that stimulate the synthesis of these compounds from secondary metabolism, could be very useful in herbicide tolerance [21]. Despite the effectiveness and low cost of most commercial safeners, some are toxic to different organisms in aquatic and terrestrial ecosystems [22]. Therefore, the search for new herbicide protective compounds that are efficient and more environmentally friendly is imperative and currently represents a new direction of research in herbicide development.

The use of biostimulants as a means for increasing the tolerance of plants when exposed to different stress factors, including herbicides, could be very positive [16,23]. Thus, the growth of plants that are under stress conditions such as drought, nutrient shortage, and other factors, including herbicides, could be improved by biostimulant application [24,25,26,27]. Surprisingly, very limited information exists about the effects of L-α-amino acid-based biostimulants on herbicide-stressed plants [28].

The R&D department of Bioiberica, S.A. company, developed an L-α-amino acid-based biostimulant called Terra-Sorb^®^. It was obtained through an exclusive enzymatic hydrolysis technology (Enzyneer^®^) of animal-origin proteins, and it contains all of the biologically active amino acids in plants. Other studies observed that the application of this biostimulant improved the regulation of Rubisco, tolerance to cold stress, and nitrogen deficiency [29,30,31].

The first objective of this study is to verify whether the combined application of an Imazamox-based herbicide with the biostimulant Terra-Sorb^®^ reduces the phytotoxic effect compared to treatment with the herbicide application alone. This herbicide-biostimulant combination was chosen given the negative effects of the herbicide on the amino acid synthesis and the significant amino acid content of the biostimulant. In addition, through the analysis of ALS activity, AAs and phenolics profiles, GSH metabolism, and stress indicators, this study tries to elucidate which mechanisms of resistance to herbicides are induced by the application of the foliar Bioiberica biostimulant (Terra-Sorb^®^).

## 2. Results and Discussion

Herbicide application is currently the most widely used and efficient agricultural practice to restrict weed proliferation. Herbicides are biologically active organic compounds that alter the basic physiological processes, causing plant death [5]. However, different studies showed that herbicides have also restricted the growth and development of plants for agronomic interests. One of the most obvious symptoms of herbicide phytotoxicity is biomass loss, which has been well characterized in numerous species and is, therefore, one of the most widely used indices to define the degree of resistance or sensitivity to this stress type [5,6]. Therefore, biomass production and RGR were measured as indicators of the protective effect of the Terra-Sorb^®^ biostimulant against herbicide application. The application of the herbicide Pulsar_40 to sunflower plants caused wilting symptoms, turgor loss, the appearance of chlorotic spots on leaves, and less growth between leaf nodes (Figure 1). However, the application of the Terra-Sorb^®^ biostimulant effectively mitigated these negative symptoms (Figure 1), and plants showed 52% more shoot biomass and almost six times more RGR compared to plants treated only with the herbicide (Figure 2).

The analysis of ALS enzyme activity is the main indicator of the phytotoxic effect of imidazolinone-type herbicides such as Imazamox (Pulsar 40) when used in sunflower plants [3,4]. In our experiment, plants supplied with the Terra-Sorb^®^ biostimulant and the herbicide registered greater ALS enzyme activities compared to plants treated only with the Pulsar 40 herbicide (Figure 3). This result indicates the protective effect of ALS activity by Terra Sorb^®^ foliar against the herbicide.

We analyzed the aminogram profile of the sunflower plants to relate the ALS activity to the amino acids profile of the sunflower plants. Thus, the lower ALS enzyme activity (Figure 3) might reduce concentrations of leucine, isoleucine, and valine in plants treated with the herbicide, as observed in other studies [32,33]. Otherwise, the sunflower plants treated with the biostimulant Terra-Sorb^®^ together with the herbicide showed higher concentrations of these amino acids, which suggests the protective effect of this biostimulant to avoid the reduction of amino acid synthesis (Table 1). This effect could have a great contribution to the best growth observed in the sunflower plants of the combined treatment (Figure 1 and Figure 2). In addition, plants supplied with Terra-Sorb^®^ and the herbicide presented higher levels of alanine and aspartate, which are precursors of leucine, isoleucine, and valine [34]. Likewise, the concentrations of other amino acids, such as glutamate, glycine, serine, and proline, were higher in plants treated with the combined treatment compared to plants treated only with the herbicide (Table 1). The increase in these amino acids might be essential to the protective effect that this compound exerts against the herbicide.

In addition to biomass production and ALS activity, the use of different photosynthetic parameters is commonly used as an indicator of the phytotoxic effects of many herbicide types [9,13,35,36]. Photosynthetic activity analysis through fluorescence is a quick, non-invasive method to check the physiological state of plants. This technique is based on the induction of fluorescence emission to a dark-adapted leaf. The fluorescence may be represented over time with a curve showing O, J, I, and P phases [10], as observed in Figure 4. In the present study, sunflower plants treated with the combined treatment showed lower fluorescence values during all the OJIP phases of fluorescence emission, lower V_J_ and DI_o_/RC, and higher S_m_ values in comparison to plants treated with the herbicide alone (Figure 4 and Figure 5). These results suggest that a lower energy dissipation, and therefore better use of light energy, is produced in plants supplied with the biostimulant under herbicide toxic conditions. Other OJIP parameters also indicated a positive influence of the Terra-Sorb^®^ application. Thus, the greater RC/ABS and ɣ_RC_ point led to a greater availability of RC and energy capture. In addition, the higher N, ψ_o_, Φ_Eo_ indicated a better quinone A reduction and electron transport capacities. Finally, the great increments in PI_ABS_ and PI_total_ showed an overall improvement in photosynthetic performance (Figure 5). In their study, Balabanova et al. [12] = observed a diminution in these parameters as a response to Imazamox in sunflower plants but did not observe a positive effect in the biostimulant application together with the herbicide compared to the herbicide alone. These results suggest different responses of fluorescence parameters in response to different biostimulants.

The application of herbicides usually produces an induction of oxidative stress. Indeed, the concentrations of indicators such as MDA and ROS tend to increase in plants affected by herbicide phytotoxicity [9]. Under these stress conditions, the application of biostimulants could reduce cellular oxidative damage and, therefore, the peroxidation of membrane lipids by regulating the antioxidant defense and reducing ROS levels in plants [37]. Accordingly, in the present experiment, sunflower plants treated with the Terra-Sorb^®^ biostimulant showed lower MDA, O_2_^−^ and H_2_O_2_ values, which confirmed the protective effect of the Terra-Sorb^®^ in reducing the phytotoxic effect of the Pulsar-40 herbicide on sunflower plants (Table 2).

One of the most important mechanisms in non-target herbicide resistance involves the action of the GSH tripeptide and the GST enzyme [38]. The combined action of the GSH-GST system produces the GSH–herbicide complex, from which the herbicide undergoes a series of transformations, thus reducing its phytotoxic effects [38,39,40]. In the present experiment, sunflower plants treated with the herbicide plus biostimulant presented greater GSH concentrations and GST activities than plants supplied with the herbicide alone. The greater GSH levels could be favored by the higher presence of glutamic, glycine, and serine (Table 1). This last amino acid produces cysteine through the enzyme serine acetyltransferase, which, together with glutamic and glycine, is one of the three amino acids that make up the GSH tripeptide [41]. Briefly, the induction of the GSH-GST in plants treated with the Terra-Sorb^®^ and Pulsar-40 herbicide could be defined as an action mechanism that would explain the protective biostimulant effect inducing the phytotransformation and/or degradation of the herbicide, and therefore reducing its intracellular concentration and its phytotoxicity in plants.

There are a lot of studies which focus on the involvement of GSH acting directly in the detoxification of toxic radicals produced by the herbicide application on plants, such as ROS and the methylglyoxal compound. This detoxification process is crucial in the so-called NTSR and is carried out by enzymes such as GPX and glyoxalases (Gly I and Gly II) [17,18]. In our experiment, plants supplied with the combined treatment showed higher GPX and Gly activities compared to the plants treated with the herbicide alone (Table 3). The induction of these enzymes in sunflower plants by the application of the Terra-Sorb^®^ could contribute to the observed minimum values of the oxidative stress indicators (Table 2).

Finally, regarding ROS detoxification, it should be noted that higher proline levels were detected in plants supplied with the biostimulant (Table 1). Proline is a very reliable indicator of resistance against abiotic stress since it has an osmoprotective, osmoregulatory, and antioxidant role against the generation of ROS [42,43]. The increase in proline concentration in the plants subjected to the combined treatment could explain the decrease in ROS levels, especially H_2_O_2_ (Table 2).

The phenolics profile was analyzed because these compounds constitute another herbicide resistance mechanism that enhances the antioxidant response of plants [21]. Moreover, the application of a phenol compound-like salicylic acid as a “safener” improves herbicide resistance in sensitive plants because it induces antioxidant plant responses [19,20]. These results have led to the proposal of the direct use of salicylic acid and other polyphenols as “safeners” or the use of compounds that stimulate their synthesis from secondary metabolism [20,21]. Additionally, we analyzed the phenylalanine ammonium-lyase (PAL) enzyme activity, which is key in the synthesis of phenolic compounds [44]. Nevertheless, neither the PAL activity nor the polyphenolic compounds which were detected varied significantly between the applied treatments (Figure 6 and Table 4). Considering that neither the PAL activity (Figure 5) nor the concentration of salicylic acid nor the rest of the phenolic compounds (Table 4) were modified with the application of Terra-Sorb^®^, we can rule out this physiological process as part of the mode of action that could be involved in the protective effect of the biostimulant.

## 3. Materials and Methods

### 3.1. Plant Material and Experimental Design

Sunflower seeds, *Helianthus annuus* var. Neoma (first generation Clearfield) were sown in individual pots (13 cm in upper diameter, 10 cm in lower diameter, 12.5 cm in height, and a volume of 2 L) filled with peat. A total of 48 plants were used for the experiment. These plants were grouped in 6 plastic trays (8 plants per tray). The sunflower plants grew in a culture chamber in the Department of Plant Physiology of the University of Granada under controlled conditions with relative humidity 60–80%, temperature 25 °C/15 °C (day/night), and 16 h/8 h photoperiod with a PPFD (photosynthetic photon-flux density) of 350 µmol^−2^ s^−1^ (measured with an SB quantum 190 sensor, LI-COR Inc., Lincoln, NE, USA). The light source was provided by fluorescent tubes (Philips Master TL-D 58W/840 REFLEX, Holland; 400–700 nm). Plants were watered with tap water until the first true leaves emerged. After, a complete Hogland-type nutrient solution was applied to the plants as irrigation composed of: 4 mM KNO_3_, 2 mM Ca(NO_3_)_2_, 2 mM MgSO_4_, 1 mM KH_2_PO_4_, 1 mM NaH_2_PO_4_, 2 µM MnCl_2_, 1 µM ZnSO_4_, 0.25 µM CuSO_4_, 0.1 µM Na_2_MoO_4_, 125 µM Fe-EDDHA, and 50 µM H_3_BO_3_ (pH 5.8). To avoid salt accumulation, the nutrient solution in the trays was removed, the trays were thoroughly cleaned using distilled water, and a new nutritive solution was added. This process was repeated every three days.

Treatments were applied 47 days after sowing and when the sunflower plants presented 6 fully expanded mature leaves. A foliar application with a volume of 0.02 mL/cm^2^ was applied to the plants of the two different treatments:Herbicide treatment (H): Pulsar_40 (4 mL/L) + Dash_HC (2 mL/L).
Herbicides + Biostimulant (HB): Pulsar_40 (4 mL/L) + Dash_HC (2 mL/L) + Terra Sorb^®^ foliar (8 mL/L).

Pulsar_40 herbicide (BASF Española, S.L., Barcelona, Spain) contains Imazamox at 4% weight/volume (*w*/*v*). The different treatments were diluted using distilled water to a final volume of 100 mL at the concentrations described above. Dash_HC (BASF Española, S.L., Barcelona) was added as a surfactant, which is composed of methyl oleate/palmitate 37.5%, fat alcohol polyalkoxylate phosphate 22.5%, and oleic acid 5%.

The foliar treatments were applied to plants in 3 trays randomly distributed in the cultivation chamber. A total of 24 plants received each of the treatments. Two samplings of the plant materials (leaves) were carried out. The first sampling was on the treatments application date in which only the fresh weight (FW) of the shoots was measured. Four days after the application of the treatments, the fluorescence of chlorophyll *a* was analyzed, and subsequently, the second sampling of the plant material was carried out. The plant material was washed and subsequently dried on filter paper to obtain the FW. From the FW data of both samplings, the relative growth rate (RGR) was calculated. A part of the plant material from the second sampling was subjected to quick freezing using liquid nitrogen and subsequently transferred to the lab for the determination of biochemical parameters. The remaining plant material was lyophilized to determine the profile of phenolic compounds. For each analysis, 3 subsamples were obtained from the frozen or lyophilized material. Then, from each of the sub-samples, the analysis was repeated in triplicate.

### 3.2. Relative Growth Rate (RGR)

The RGR was calculated using the increase in FW of the plants from the moment of the application of treatments to the moment of sampling using the equation:RGR = (ln FW_f_ − ln FW_i_)/(T_f_ − T_i_)

T is the time (number of days), and the subscripts indicate the first (i) and last sampling (f) [45].

### 3.3. Determination of Enzymatic Activities

The assay method for the acetolactate synthase (ALS) enzyme was carried out following the method described by Malkawi et al. [46], in which the acetolactate reaction product was measured spectrophotometrically at an absorbance of 525 nm.

To determine the enzyme activities related to GSH homeostasis [glutathione S-transferase, glutathione peroxidase (GPX), glyoxalase I (Gly I), and glyoxalase II (Gly II)], the enzyme extracts were obtained by grinding 0.5 g of leaves in 5 mL of 50 mM phosphate buffer (pH 7.0) containing 100 mM KCl, 5 mM mercaptoethanol, and 10% glycerol. The homogenate was centrifuged at 21,500× *g* for 10 min, and the supernatant was used to determine the enzymatic activities [47].

Glutathione S-transferase (GST) activity was measured according to Hasanuzzaman et al. [47], where the reaction mixture contained 1.5 mM GSH, 1 mM 1-chloro-2,4-dinitrobenzene (CDNB), and enzyme extract. The increase in absorbance at 340 nm was recorded. GPX activity was measured as described by Nahar et al. [48]. The absorbance was measured at 412 nm in a maximum time of 5 min. Gly I activity was analyzed according to the method of Hasanuzzaman et al. [47] and modified as follows: after adding methylglyoxal, the reaction started, and the increase in the absorbance at 240 nm was measured. Gly II activity was analyzed as described by Hasanuzzaman et al. [47]. The reaction was started by adding S-lactoylglutathione, and the increase in the absorbance at 412 nm was recorded spectrophotometrically.

The phenylalanine ammonium lyase (PAL) activity was analyzed according to Rivero et al. [49] by a method which measured the production of cinnamic acid at 290 nm.

### 3.4. Aminogram Analysis

The soluble amino acids were extracted following the method of Bieleski and Turner [50] modified as follows: 0.1 g of fresh leaves were homogenized in 1 mL of MCW (methanol:chloroform:water, 12:5:1). An amount of 50 µL of L-2 aminobutyric acid was added as an internal standard. The mixture was centrifuged at 2300× *g* for 10 min. An amount of 700 µL of Milli-Q water and 1.2 mL of chloroform were added to the resulting supernatant and incubated for 24 h at 4 °C. Then, the aqueous phase was obtained, which was lyophilized, and the resulting dry extract was diluted with 0.1 M HCl. The instrumental analysis of soluble amino acids was carried out using the Pre-column AccQ Tag Ultra Derivatization Kit (Waters, Milford, MA, USA). Derivatization was carried out according to the manufacturer’s protocol. For derivatization, 60 µL of borate buffer was added to 10 µL of the sample, 10 µL of 0.1 N NaOH, and 20 µL of reconstituted AccQ Tag Ultra reagent. LC fluorescence analysis was performed on the Waters Acquity^®^ UHPLC system equipped with the Acquity fluorescence detector. UHPLC separation was performed on the AccQ Tag Ultra column (2.1 × 100 mm, 1.7 µm) from Waters. The flow rate was 0.7 mL min^−1^, and the column temperature was kept at 55 °C. The injection volume was 1 µL, and detection was established at an excitation wavelength of 266 nm and an emission wavelength of 473 nm. The solvent system consisted of two eluents: 1:20 dilution of concentrated AccQ Tag Ultra eluent A and AccQ Tag Ultra eluent B.

### 3.5. Chl a Fluorescence

The analysis was performed on intact plants. For each measurement, a small area of the leaf was kept in darkness using a special clip. Chl *a* fluorescence kinetics was determined using the Handy PEA Chlorophyll Fluorimeter (Hansatech Ltd., King’s Lynn, Norfolk, UK); OJIP phases were induced by red light (650 nm) with a light intensity of 3000 µmol photons m^−2^ s^−1^. One measurement was made on nine fully developed leaves from nine different plants from each treatment (*n* = 9). The analyzed leaves were in the middle position of the plant. This analysis was performed at midday The phases of the OJIP fluorescence were analyzed by the JIP test and the following parameters were obtained: time to reach fluorescence maximum (t(F_M_)), fluorescence origin (F_O_), fluorescence maximum (F_M_), variable fluorescence (F_V_), maximum quantum yield of primary PSII photochemistry (F_V_/F_M_), fluorescence at J-step (2 ms) (V_J_), standardized area above the fluorescence curve (S_m_), the number of quinone A (Q_A_) redox turnovers until F_M_ was reached (N), the apparent antenna size of active photosystem II (PSII) reaction center (RC) (ABS/RC), proportion of active RC (RC/ABS), dissipated energy flux per RC at t = 0 (DI_o_/RC), trapping flux leading to Q_A_ reduction per RC (TR_o_/RC), electron transport flux per RC at t = 0 (ET_o_/RC), the efficiency that a trapped exciton can move an electron further than Q_A_ into the electron transport chain (ψ_o_), quantum yield for electron transport from Q_A_ to plastoquinone (φ_Eo_), the probability that the PSII Chl molecule functions as RC (ɣ_RC_), performance index of electron flux from PSII-based to intersystem acceptors (PI_ABS_), and performance index of electron flux to the final photosystem I (PSI) electron acceptors (PI_total_) [51]. The results of these parameters are shown in Figure 5. The represented data were obtained by dividing the mean of H + B treatment plants by the mean of H treatment plants for each parameter to normalize the data and enable the comparison of parameters of different scales.

### 3.6. Determination of the Concentration of Oxidative Indicators (Malondialdehyde (MDA), H_2_O_2_, and O_2_^−^)

MDA concentration was determined according to the method described by Fu and Huang [52]. Fresh materials were extracted with TBA + TCA, and after extraction, the absorbance was recorded at 532 nm and 600 nm to the correct turbidity. H_2_O_2_ concentration was measured colorimetrically according to Junglee et al. [53] based on the reaction with KI and reading absorbance at 350 nm. For O_2_^−^ determination, the method described by Xiao et al. [54] was followed. The method was based on the reaction of the sample extract with hydroxylamine, sulfanilic acid, and α-1-naphthylamine, and the color intensity was measured at 530 nm.

### 3.7. Determination of GSH Concentration

The determination of GSH concentration was carried out following the method of Law et al. [55]. This method is based on the specificity of the enzyme GSH reductase for oxidized glutathione. Finally, the samples were read at 412 nm against a GSH standard curve.

### 3.8. Determination of the Phenolic Compound Profile

Dry lyophilized leaves (50 mg) were extracted with 1 mL of methanol 70% *v*/*v* in a vortex for 1 min; it was then heated to 70 °C for 30 min in a heat bath, stirring every 5 min using a vortex and centrifuged at 12,000× *g* for 10 min at 4 °C. The supernatant was collected, and the methanol was completely removed using a rotary steam. The dried material obtained was redissolved in 1 mL of ultrapure water and filtered through a 0.22-micron Millex-HV13 filter (Millipore, Billerica, MA, USA). Phenolic compounds were determined using a high-performance ion-exchange liquid chromatography method which separated the phenolic compounds according to the procedure of Moreno et al. [56]. First, the separated phenolic compounds were identified from the extracted samples following their MS^2^-[MH] fragments in HPLC-DAD-ESI-MSN, carried out on a Luna C18 100A column (250 × 4.6 mm, 5 microns in particle size; Phenomenex, Macclesfield, UK). For mobile phases A and B, respectively, water was used: formic acid (99:1, *v*/*v*) and acetonitrile A and B, with a flow rate of 800 µL/min. The linear gradient started with 1% solvent B, reaching 17% solvent B in 15 min to 17 min, 25% at 22, 35% at 30, and 50% at 35, which was kept in isocratic mode up to 45 min. Chromatograms were recorded at 330 nm. HPLC-DAD-ESI analysis was carried out on an Agilent 1200 HPLC (Agilent Technologies, Waldbronn, Germany) coupled to a serial mass detector.

### 3.9. Statistical Analysis

The results were statistically evaluated using an analysis of variance, and simple ANOVA with a 95% confidence interval. For all parameters analyzed, the mean and standard error were calculated from nine data (*n* = 9). Differences between treatment means were compared using Fisher’s least significant differences (LSD) test at a 95% probability level. The significance levels were expressed as: * *p* < 0.05; ** *p* < 0.01; *** *p* < 0.001; NS—not significant.

## 4. Conclusions

In conclusion, the application of the Terra-Sorb^®^ biostimulant shows a protective effect against stress due to the imazamox-based Pulsar-40 herbicide. Thus, the combined treatment incremented plant growth and ALS enzymatic activity and maintained leucine, isoleucine, and valine concentrations. In addition, the application of the biostimulant combined with the imazamox herbicide protected photosynthetic activity and significantly reduced oxidative stress in the sunflower plants. This protective effect could be based on the induction of the GSH-GST and antioxidant enzymatic systems. In addition, a higher accumulation of proline could substantially reduce the radical and toxic reactive compounds produced by herbicide application.

## Figures and Tables

**Figure 1 plants-11-02761-f001:**
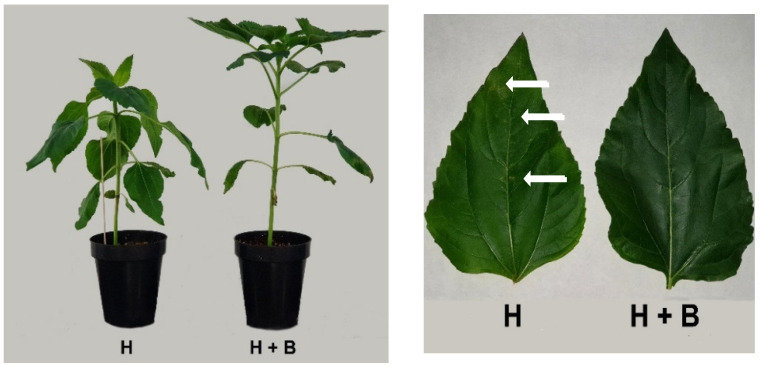
Sunflower plants and leaves 4-days after the application of the treatments (H: Pulsar_40; H + B: Pulsar_40 + Terra-Sorb^®^). Arrows on sunflower leaves indicate chlorotic spots on plants treated with the herbicide (H).

**Figure 2 plants-11-02761-f002:**
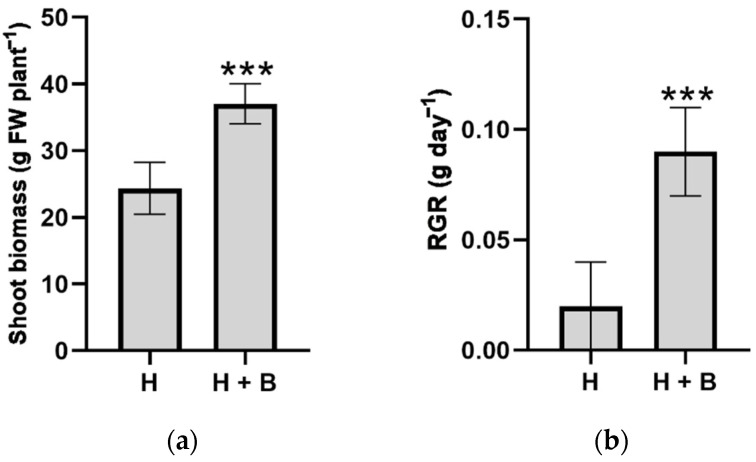
Biomass production (**a**) and relative growth rate (RGR) (**b**) of shoot of sunflower plants subjected to the treatments (H: Pulsar 40; H + B: Pulsar 40 + Terra-Sorb^®^). Values are expressed as means ± standard error (*n* = 9). The level of significance was *p* < 0.001 (***).

**Figure 3 plants-11-02761-f003:**
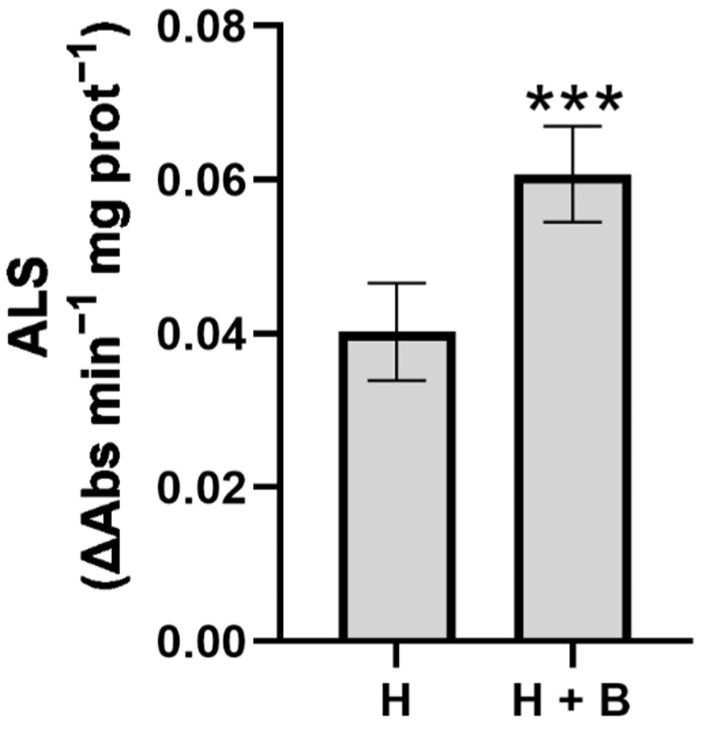
ALS activity in sunflower plants subjected to the treatments (H: Pulsar-40; H + B: Pulsar-40 + Terra Sorb^®^ foliar). Values are expressed as means ± standard error (*n* = 9). The level of significance was represented by *p* < 0.001 (***).

**Figure 4 plants-11-02761-f004:**
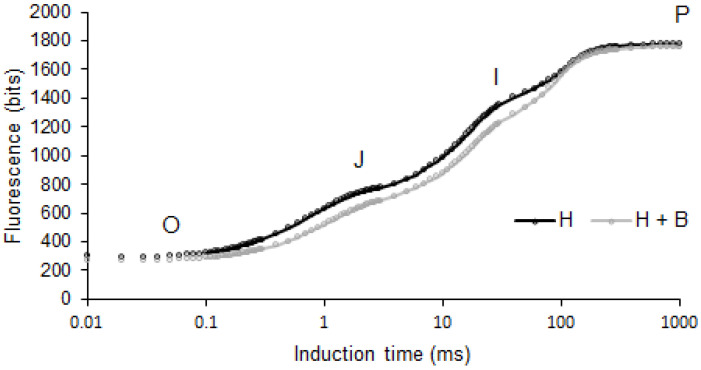
Native fluorescence induction curves of sunflower plants subjected to the treatments (H: Pulsar-40; H + B: Pulsar-40 + Terra Sorb^®^ foliar). Represented values are means from nine data (*n* = 9).

**Figure 5 plants-11-02761-f005:**
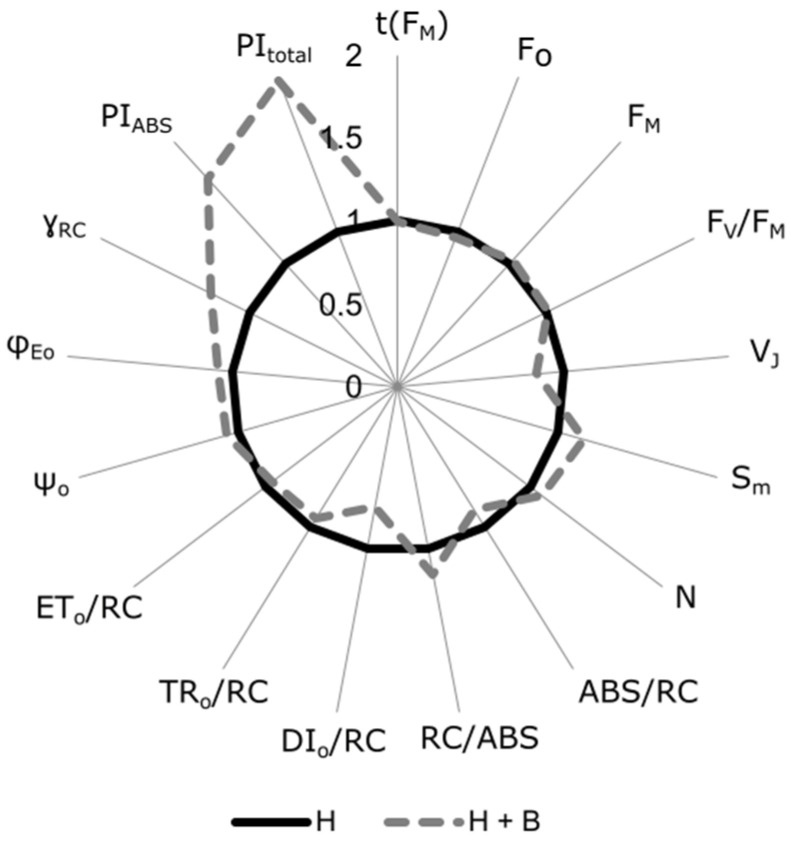
Radar plot showing OJIP-test parameters derived from Chl *a* fluorescence induction in sunflower plants subjected to H + B treatment: Pulsar-40 + Terra Sorb^®^ foliar. Data normalization were made to enable the representation of all parameters on the same scale. The dotted gray line values represent the relative increase or decrease in plants of the H + B treatment with respect to plants of the H treatment (black line).

**Figure 6 plants-11-02761-f006:**
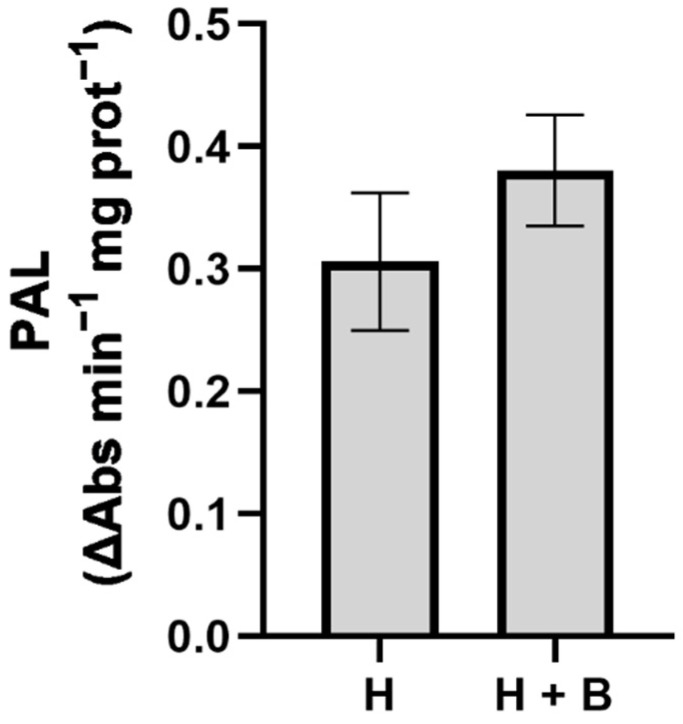
PAL activity in sunflower plants subjected to the treatments (H: Pulsar-40; H + B: Pulsar-40 + Terra-Sorb^®^). Values are expressed as means ± standard error (*n* = 9).

**Table 1 plants-11-02761-t001:** Aminogram (µg g^−1^ DW) in sunflower plants subjected to the treatments.

	H	H + B	*p*-Value
Alanine	145.69 ± 2.30	210.92 ± 3.33	***
Arginine	37.26 ± 0.59	36.13 ± 0.57	NS
Asparagine	ND	ND	
Aspartate	237.33 ± 2.08	306.49 ± 4.85	***
Glutamate	381.49 ± 6.03	430.49 ± 6.81	***
Glutamine	ND	ND	
Glycine	169.06 ± 2.67	195.13 ± 3.09	***
Histidine	11.72 ± 0.22	11.49 ± 0.21	NS
Isoleucine	460.64 ± 7.28	751.21 ± 11.88	***
Leucine	821.21 ± 12.98	1113.13 ± 17.60	***
Lysine	7.23 ± 0.11	7.30 ± 0.12	NS
Phenylalanine	479.71 ± 7.58	485.64 ± 7.68	NS
Proline	43.20 ± 0.68	62.93 ± 1.00	***
Serine	635.08 ± 10.04	691.70 ± 10.94	***
Threonine	42.86 ± 0.68	42.63 ± 0.67	NS
Tryptophan	355.27 ± 5.62	361.70 ± 5.72	NS
Tyrosine	512.92 ± 8.11	514.62 ± 8.14	NS
Valine	1008.64 ± 15.95	1475.79 ± 23.33	***
Methionine	75.51 ± 1.19	75.88 ± 1.20	NS
Cysteine	7.39 ± 0.13	7.23 ± 0.12	NS

H: Pulsar-40; H + B: Pulsar-40 + Terra-Sorb^®^. Values are means ± standard error (*n* = 9). The levels of significance were represented as NS (*p* > 0.05) and *** (*p* < 0.001). ND—Non detected.

**Table 2 plants-11-02761-t002:** Stress indicators in sunflower plants subjected to the treatments.

	MDA (μg g^−1^ FW)	O_2_^−^ (μg g^−1^ FW)	H_2_O_2_ (μg g^−1^ FW)
H	9.68 ± 0.49	1.64 ± 0.35	12.64 ± 1.47
H + B	8.41 ± 0.35	1.08 ± 0.27	8.33 ± 1.81
*p*-value	***	*	***

H: Pulsar-40; H + B: Pulsar 40 +Terra-Sorb^®^. Values are means ± standard error (*n* = 9). The levels of significance were represented as * (*p* < 0.05) and *** (*p* < 0.001).

**Table 3 plants-11-02761-t003:** GSH homeostasis in sunflower plants subjected to the treatments.

	GSH (mg g^−1^ FW)	GST (ΔAbs min^−1^ mg prot^−1^)	GPX (ΔAbs min^−1^ mg prot^−1^)	Gly I (ΔAbs min^−1^ mg prot^−1^)	Gly II (ΔAbs min^−1^ mg prot^−1^)
H	0.14 ± 0.02	0.32 ± 0.07	0.04 ± 0.01	0.06 ± 0.01	0.05 ± 0.01
H + B	0.27 ± 0.05	1.04 ± 0.27	0.07 ± 0.01	0.15 ± 0.03	0.10 ± 0.02
*p*-value	***	**	***	***	***

H: Pulsar_40; H + B: Pulsar_40 +Terra-Sorb^®^. Values are means ± standard error (*n* = 9). The levels of significance were represented as ** (*p* < 0.01) and *** (*p* < 0.001).

**Table 4 plants-11-02761-t004:** Phenolic compound profile (mg g^−1^ DW) in sunflower plants subjected to the treatments.

	H	H + B	*p*-Value
5-CQA (chlorogenic acid)	3.17 ± 0.10	3.17 ± 0.13	NS
5-CQA-Isomer	4.03 ± 0.05	4.03 ± 0.05	NS
Di-caffeoylquinic acid	10.56 ± 2.10	10.56 ± 2.02	NS
SA (µg g^−1^ DW)	4.03 ± 0.06	4.07 ± 0.06	NS

H: Pulsar_40; H + B: Pulsar_40 + Terra-Sorb^®^. Values are means ± standard error (*n* = 9). The level of significance is represented as NS (*p* > 0.05).

## Data Availability

Not applicable.

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
