# Peer review of "Application of an Enzymatic Hydrolysed L-α-Amino Acid Based Biostimulant to Improve Sunflower Tolerance to Imazamox"

_plants, 2022, doi:10.3390/plants11202761_

Round 1
Reviewer 1 Report
The experiment was well performed and the theme is novel and very interesting, but some improvements are needed. Overall, the manuscript will meet the publishing standard of the journal after revisions.
Abstract:
There is an imbalance between the parts of the abstract as the introductory sentences represents more than one third of the abstract (lines 10-14) and the other parts (objectives - materials and methods - results - conclusion) represent less than two thirds of the abstract (lines 15-24). I think the parts of the abstract have to be balanced.
Introduction:
1. Lines 78-80: Please add citations to the sentence. I suggest using the following articles as citations to the sentence:
- Desoky, E.M., Elrys, A.S., Mansour, E., Eid, R.S.M., Selem, E., Ali, E.F., Mersal, G.A.M., Rady, M.M., Semida, W.M. (2021). Application of biostimulants promotes growth and productivity by fortifying the antioxidant machinery and suppressing oxidative stress in faba bean under various abiotic stresses. Scientia Horticulturae, 288: 110340.
- Alharby, H.F., Alzahrani, Y., Rady, M.M. (2020). Seeds pretreatment with zeatins or maize grain-derived organic biostimulant improved hormonal contents, polyamine gene expression, and salinity and drought tolerance of wheat. International Journal of Agriculture and Biology, 24(4): 714-724.
- Taha, R.S., Alharby, H.F., Bamagoos, A.A., Medani, R.A., Rady, M.M. (2020). Elevating tolerance of drought stress in Ocimum basilicum using pollen grains extract; a natural biostimulant by regulation of plant performance and antioxidant defense system. South African Journal of Botany, 128: 42-53.
- Rehman, H., Alharby, H.F., Alzahrani, Y., Rady, M.M. (2018). Magnesium and organic biostimulant integrative application induces physiological and biochemical changes in sunflower plants and its harvested progeny on sandy soil. Plant Physiology and Biochemistry, 126: 97–105.
Results and Discussion:
2. Lines 103 and 105: The authors wrote the full name and the abbreviation in parentheses for the two treatments applied in the study. After that, there is no need to write the full name, but the abbreviations must be used until the end of this part. Then the full names should be written in the conclusions, if any.
3. Line 120: Please write the full name of ALS enzyme for the first time.
4. Table 2: Please change “O2.-” to “O2•â€’”. Also, it is best to standardize the units of measurement for MDA, H2O2, and O2•â€’.
Materials & Methods:
5. Lines 273, 275, and 277: The authors wrote mL and ml, please unify throughout this part and other parts if any.
For treatments, Isn't it more feasible to have a control treatment in which the herbicide and/or biostimulant are not applied? Also, could the biostimulant be tested alone in a fourth treatment?
Author Response
The experiment was well performed and the theme is novel and very interesting, but some improvements are needed. Overall, the manuscript will meet the publishing standard of the journal after revisions.
Abstract:
There is an imbalance between the parts of the abstract as the introductory sentences represents more than one third of the abstract (lines 10-14) and the other parts (objectives - materials and methods - results - conclusion) represent less than two thirds of the abstract (lines 15-24). I think the parts of the abstract have to be balanced.
Response: The abstract was modified to balance the different parts
Introduction:
- Lines 78-80: Please add citations to the sentence. I suggest using the following articles as citations to the sentence:
- Desoky, E.M., Elrys, A.S., Mansour, E., Eid, R.S.M., Selem, E., Ali, E.F., Mersal, G.A.M., Rady, M.M., Semida, W.M. (2021). Application of biostimulants promotes growth and productivity by fortifying the antioxidant machinery and suppressing oxidative stress in faba bean under various abiotic stresses. Scientia Horticulturae, 288: 110340.
- Alharby, H.F., Alzahrani, Y., Rady, M.M. (2020). Seeds pretreatment with zeatins or maize grain-derived organic biostimulant improved hormonal contents, polyamine gene expression, and salinity and drought tolerance of wheat. International Journal of Agriculture and Biology, 24(4): 714-724.
- Taha, R.S., Alharby, H.F., Bamagoos, A.A., Medani, R.A., Rady, M.M. (2020). Elevating tolerance of drought stress in Ocimum basilicum using pollen grains extract; a natural biostimulant by regulation of plant performance and antioxidant defense system. South African Journal of Botany, 128: 42-53.
- Rehman, H., Alharby, H.F., Alzahrani, Y., Rady, M.M. (2018). Magnesium and organic biostimulant integrative application induces physiological and biochemical changes in sunflower plants and its harvested progeny on sandy soil. Plant Physiology and Biochemistry, 126: 97–105.
Response: These references were added. L84
Results and Discussion:
- Lines 103 and 105: The authors wrote the full name and the abbreviation in parentheses for the two treatments applied in the study. After that, there is no need to write the full name, but the abbreviations must be used until the end of this part. Then the full names should be written in the conclusions, if any.
Response: Thank you for the comment. The abbreviations were deleted because we think this improves the readability of the text.
- Line 120: Please write the full name of ALS enzyme for the first time.
Response: Done. L38
- Table 2: Please change “O2.-” to “O2•â€’”. Also, it is best to standardize the units of measurement for MDA, H2O2, and O2•â€’.
Response: Done. L187
Materials & Methods:
- Lines 273, 275, and 277: The authors wrote mL and ml, please unify throughout this part and other parts if any.
Response: Done. mL was replaced by ml in the manuscript
For treatments, Isn't it more feasible to have a control treatment in which the herbicide and/or biostimulant are not applied? Also, could the biostimulant be tested alone in a fourth treatment?
Response: Thank you for your comment. The experimental design was carried out to simulate field growing conditions with plants to which herbicide is applied and plants to which herbicide is applied along with the biostimulant. The objective of the experiment was to compare only these two treatments to see if the application of the biostimulant has positive effects.
Reviewer 2 Report
GENERAL Comments
The authors carried out an interesting research work on the interaction between an Imazamox based herbicide treatment and a biostimulant used as a herbicide safener. This research may be ascribed to the applied plant science area, therefore it falls within the scope of this journal.
A number of papers have already been published on related subjects, so this report may not be considered a novelty.
My main concern is the lack of proper Control treatments, since the comparison is made between two treatments only, i.e. “herbicide treated” plants compared with “herbicide + biostimulant treated” plants.
Based on this minimal experimental design, in my opinion this manuscript may be published without these Controls but it should be considered as a preliminary report. An Untreated and a Biostimulant-only treatment would greatly improve this work.
As is, this report falls into the applied plant science area but it does not allow to investigate the physiological basis of the plant response to the treatments, so the authors should make this clear in text.
Overall, the Introduction should be improved with a description of the herbicide structure, metabolism and effect s on plant physiology. The nature, composition and documented effects of the biostimulant used in this research should also be presented. The authors should explain why they chose this particular combination of herbicide + biostimulant.
The chlorophyll fluorescence technique should be introduced more clearly in order to be useful to every reader. This paragraph must be greatly improved. Reference is made to the OJIP kinetics but no data are presented, therefore there is no need to mention the JIP test parameters.
About this point I have to ask a question: According to authors OJIP kinetics were recorded, therefore why no JIP-test data are presented?
These would add a lot more information and allow for a better understanding of the physiological effects of the treatments, thus substantially improving the manuscript.
I would recommend to include these data.
The Materials and Methods section must be improved.
Detailed info about cultivation is needed (see below).
Sample collection and processing should be clearly reported (particularly the extraction procedure for sensitive metabolites that need immediate freezing). The number of replicate plants and the number of replicate extractions (rather than replicate assays of the same extract) for each determination should be clearly stated.
For Chl a fluorescence measurements it is very important to report how many of replicate measurements were recorded for each leaf/plant/treatment detailed procedure and time of the measurements.
Were the fluorescence measurements performed on the intact plants or on detached leaves?
At what time ? This is very important because the non photochemical quenching (NPQ) is gradually activated during the light period and it affects Fv/Fm, so it is important to know at what time during the light period the measurements were taken.
Results presentation should also be improved and legends to Figs and Tabs should be more complete and explicative.
The english language needs some revisions.
Following are some detailed comments, referring to the line numbering of the manuscript:
ABSTRACT
15 CHECK the sentence “enzymatically amino acid based biostimulant”: something is missing. Please explain precisely.
19 improve ? I’d suggest to use unambiguous terms, e.g. “increase” or “decrease”.
INTRODUCTION
46 - 30% of … what ?
64 - Introduce phenolics
76 - 78 Rephrase. Introduce the use of biostimulants as a means for etc…
84 - specificy what particular herbicide and why that one was chosen.
Introduce the mode of action, metabolic effect etc.. of Imazamox herbicides
This point should not be overlooked if attempting to investigate the protective action of safeners from a physiological point of view.
85 - should be improved with clear definition: enzymatic hydrolysis of what ?
88 - state, list. Avoid ambiguous terms
RESULTS & DISCUSSION
96-102 Rephrase “are reliable indicators” ?? maybe “… these parameters were measured as indicators of …”
107-120 Proper CONTROLS ARE MISSING (see general comments).
117: rephrase “level of significance was p<0.001”
120-125 Move to Introduction section. The mode of action of this herbicide should be clearly described in the introduction.
126-130 Control is missing, see comment to lines 107-120
139-143 PROBABLY ? This should be rephrased and supported with biblio references.
The whole sentence is based on the assumption that H treatment decreases ALS enzyme activity. This was certainly reported in the literature (references to be added) but it should be checked in this work, with a proper control in these particular set of experimental/environmental conditions.
148-150 this should be presented as speculations
159 besides to …check english usage
159-165 Re-phrase and improve. More info about Fv/Fm, meaning etc…
167-169 “Fv/Fm value lower than 0.85” . A definite value should be given.
Moreover, I do not think that definitive conclusions may be drawn based on these valus, unless the measurement procedure allows for that discussion. See comment above about fluorescence measurements. Authors should explain.
173 check language
Why no OJIP kinetics and/or JIP-test data are reported ? see comment above.
189-190 legend to tab 2, number of replicates? on how many different plants? what is the number +/- maybe sd should be stated, what are the asterisks? What stat test ? check english: “the levels of significance…”
192 herbicide detoxification: does this apply in the case of IMAZAMOX ? Supporting biblio refs ?
196 H+T compared to H only. These data do not allow for separate discussion
201-204 As above, treatment was H+T not T alone, therefore conclusions may only refer to the combined effect, no data on T treatment on its own. Discuss accordingly.
231 check english
Materials and Methods
263 - what is the light source ? spectrum ?
265 - Important detail about plant cultivation are missing: was the nutrient solution added as irrigation? Were the pots/plants always irrigated with the nutrient solution ? How about build up increasing salt concentration in the substrate ? Are electrical conductivity of the potting substrate data available
(beginning and end of cultivation)? Maybe some salt stress ?
270 -treatments were applied 47 days after sowing
276 - 278 - in M&M details about Pulsar and Dash reagents should be given (producer, composition)
280 - what are these “trays” ? plants were potted or not ? Please explain clearly, maybe plants were grouped in trays ? It should be clear to the reader the size of the experiment: therefore how many plants per treatment ?
284 - Details should be added: instrument, method, dark adaptation, flash intensity etc…
Correct definition for Fv/Fm ( e.g. Fv/Fm or the max efficiency of photochemistry in the DAS)
286 - Decontaminated from ?
NOTE: no DW data ? No water content ? Why ?
288-290 SAMPLING TIMING PROCESSING: explain clearly the sampling and extraction steps
This is confusing and it should be improved
Following paragraphs should be ordered according to the workflow: sampling, transfer to lab, extraction, assay …
Although apparently aminoacid composition and phenolics were assayed from the same subsample (see lines 288-290), different extraction methods are reported. see 327 and 374 .
Please explain ?
301 - Missing biblio ref Malawi et al 2003
306 - correct: grinding
306-307 5 ml final volume, P-buffer containing 100 mM KCl etc… Or were they added to the 5 mL ?
321 - merge the two short sentences
327 - according to the method… modified as follows
328 - v/v ? specify
332 - dry extract dissolved
345 - no analysis only single parameter, therefore better just chl a fluorescence
349 - OJIP was not introduced? explain or leave out.
350 - rephrase
NOTE introduction needed about Chl fluorescence, basic intro, parameters
352 - This is not correct. Energy fluxes relate to a specific set of jip-test parameters (energy cascade), while fluorescence is related to photochemistry rather than photosynthetic activity (as CO2 fixation). However, this manuscript only reports measured F0 and Fm parameters, with no OJIP kinetics and related JIP-test.
Importantly: how many replicate measurements, on how many plants /leaves? Time of day ?
360 - Timing ? Quick freezing and processing ?
According to the stat anal paragraph, 3 determinations were made for each single parameter:
3 reps may not be enough, generally speaking. But does this mean 3 replicate assays on a single extract ? Or is it 3 different extracts?
362 - missing biblio ref
362-364 - The cited method reference, points to another paper by Cubi 2008 and it is based on NBT.
Please check, explain and correct accordingly.
Morever this protocol starts from quick liquid N2 freezing, please describe the workflow clearly.
I cannot see where the mentioned reagents (hydroxylamine sulfanilic acid and α-1-naphthylamine) are used.
Check the text and the method.
CONCLUSION
402-403 Authors should only refer to the “T+H” treatment since no “T” alone treatment was included in the experimental design. In this case ”reduction” caused by H treatment or is it increase in the case of “T+H” ? Proper Control missing, therefore text should be modified accordingly.
404-409 check language
Author Response
GENERAL Comments
The authors carried out an interesting research work on the interaction between an Imazamox based herbicide treatment and a biostimulant used as a herbicide safener. This research may be ascribed to the applied plant science area, therefore it falls within the scope of this journal.
A number of papers have already been published on related subjects, so this report may not be considered a novelty.
My main concern is the lack of proper Control treatments, since the comparison is made between two treatments only, i.e. “herbicide treated” plants compared with “herbicide + biostimulant treated” plants.
Based on this minimal experimental design, in my opinion this manuscript may be published without these Controls but it should be considered as a preliminary report. An Untreated and a Biostimulant-only treatment would greatly improve this work.
As is, this report falls into the applied plant science area but it does not allow to investigate the physiological basis of the plant response to the treatments, so the authors should make this clear in text.
Overall, the Introduction should be improved with a description of the herbicide structure, metabolism and effect s on plant physiology. The nature, composition and documented effects of the biostimulant used in this research should also be presented. The authors should explain why they chose this particular combination of herbicide + biostimulant.
The chlorophyll fluorescence technique should be introduced more clearly in order to be useful to every reader. This paragraph must be greatly improved. Reference is made to the OJIP kinetics but no data are presented, therefore there is no need to mention the JIP test parameters.
About this point I have to ask a question: According to authors OJIP kinetics were recorded, therefore why no JIP-test data are presented?
These would add a lot more information and allow for a better understanding of the physiological effects of the treatments, thus substantially improving the manuscript.
I would recommend to include these data.
The Materials and Methods section must be improved.
Detailed info about cultivation is needed (see below).
Sample collection and processing should be clearly reported (particularly the extraction procedure for sensitive metabolites that need immediate freezing). The number of replicate plants and the number of replicate extractions (rather than replicate assays of the same extract) for each determination should be clearly stated.
For Chl a fluorescence measurements it is very important to report how many of replicate measurements were recorded for each leaf/plant/treatment detailed procedure and time of the measurements.
Were the fluorescence measurements performed on the intact plants or on detached leaves?
At what time ? This is very important because the non photochemical quenching (NPQ) is gradually activated during the light period and it affects Fv/Fm, so it is important to know at what time during the light period the measurements were taken.
Results presentation should also be improved and legends to Figs and Tabs should be more complete and explicative.
The english language needs some revisions.
Response: Thank you for your comment. The experimental design was carried out to simulate field growing conditions with plants to which herbicide is applied and plants to which herbicide is applied along with the biostimulant. The objective of the experiment was to compare only these two treatments to see if the application of the biostimulant has positive effects.
Several parts of the discussion have been rewritten to make it more in accordance with the experimental design used.
The information requested was added to the manuscript
The English was checked and several corrections were made
Following are some detailed comments, referring to the line numbering of the manuscript:
ABSTRACT
15 CHECK the sentence “enzymatically amino acid based biostimulant”: something is missing. Please explain precisely.
Response: Done. The word “enzymatically” was deleted to improve clarity. L15
19 improve ? I’d suggest to use unambiguous terms, e.g. “increase” or “decrease”.
Response: This word was deleted. L18
INTRODUCTION
46 - 30% of … what ?
Response: 30% of the global herbicide sales. This sentence was completed. L51-52
64 - Introduce phenolics
Response: Done. L68
76 - 78 Rephrase. Introduce the use of biostimulants as a means for etc…
Response: Done. L81-82
84 - specificy what particular herbicide and why that one was chosen.
Response: Done. L35-39 and L94-96
Introduce the mode of action, metabolic effect etc.. of Imazamox herbicides
This point should not be overlooked if attempting to investigate the protective action of safeners from a physiological point of view.
Response: Done. L35-39
85 - should be improved with clear definition: enzymatic hydrolysis of what ?
Response: Hydrolysis of animal origin proteins. The sentence was modified to improve clarity. L88
88 - state, list. Avoid ambiguous terms
Response: Done. L97-98
RESULTS & DISCUSSION
96-102 Rephrase “are reliable indicators” ?? maybe “… these parameters were measured as indicators of …”
Response: Done. L111
107-120 Proper CONTROLS ARE MISSING (see general comments).
Response: No controls were included because the objective of the experiment was to compare only these two treatments to see if the application of the biostimulant has positive effects.
117: rephrase “level of significance was p<0.001”
Response: Done. L127-128
120-125 Move to Introduction section. The mode of action of this herbicide should be clearly described in the introduction.
Response: Done. L36-39
126-130 Control is missing, see comment to lines 107-120
Response: This part was rewritten to make it more in accordance with the experimental design. L132-134
139-143 PROBABLY ? This should be rephrased and supported with biblio references.
The whole sentence is based on the assumption that H treatment decreases ALS enzyme activity. This was certainly reported in the literature (references to be added) but it should be checked in this work, with a proper control in these particular set of experimental/environmental conditions.
Response: This part was modified to improve accuracy and references were added. L143-145
148-150 this should be presented as speculations
Response: Done. L152-154
159 besides to …check english usage
Response: Done. L164
159-165 Re-phrase and improve. More info about Fv/Fm, meaning etc…
167-169 “Fv/Fm value lower than 0.85” . A definite value should be given.
Moreover, I do not think that definitive conclusions may be drawn based on these valus, unless the measurement procedure allows for that discussion. See comment above about fluorescence measurements. Authors should explain.
Response: This part was rewritten and more information about Fv/Fm and Chl a fluorescence was added. L167-177
173 check language
Response: Done. L197
Why no OJIP kinetics and/or JIP-test data are reported ? see comment above.
Response: OJIP kinetics was added in Figure 4. L182-184
189-190 legend to tab 2, number of replicates? on how many different plants? what is the number +/- maybe sd should be stated, what are the asterisks? What stat test ? check english: “the levels of significance…”
Response: The missing information was added to the legend and English was improved. L193-194
192 herbicide detoxification: does this apply in the case of IMAZAMOX ? Supporting biblio refs ?
Response: Yes, this applies in the case of IMAZAMOX. Reference was added. L209
196 H+T compared to H only. These data do not allow for separate discussion
Response: This sentence was modified to better discuss. L212
201-204 As above, treatment was H+T not T alone, therefore conclusions may only refer to the combined effect, no data on T treatment on its own. Discuss accordingly.
Response: This part was rewritten to make it more in accordance with the experimental design. L217-218
231 check english
Response: Done. L247-248
Materials and Methods
263 - what is the light source ? spectrum ?
Response: This information was added. The light source was provided by fluorescent tubes (Philips Master TL-D 58W/840 RE-FLEX, Holland). Spectrum: 400-700 nm. L283-284
265 - Important detail about plant cultivation are missing: was the nutrient solution added as irrigation? Were the pots/plants always irrigated with the nutrient solution ? How about build up increasing salt concentration in the substrate ? Are electrical conductivity of the potting substrate data available
(beginning and end of cultivation)? Maybe some salt stress ?
Response: Missing information about plant cultivation was added. No data about electrical conductivity of the potting substrate is available. L284-289
270 -treatments were applied 47 days after sowing
Response: Corrected. L291
276 - 278 - in M&M details about Pulsar and Dash reagents should be given (producer, composition)
Response: Done. L297-301
280 - what are these “trays” ? plants were potted or not ? Please explain clearly, maybe plants were grouped in trays ? It should be clear to the reader the size of the experiment: therefore how many plants per treatment ?
Response: Plants were grown in pots and these pots were grouped in plastic trays. 24 plants per treatment. This information was added in the text. L278-278 and L303-304
284 - Details should be added: instrument, method, dark adaptation, flash intensity etc…
Correct definition for Fv/Fm ( e.g. Fv/Fm or the max efficiency of photochemistry in the DAS)
Response: This information is specified after in Chl a fluorescence subsection. The definition for Fv/Fm was corrected. L370-377
286 - Decontaminated from ?
Response: This was a mistake, the word “decontaminated” was replaced. L309
NOTE: no DW data ? No water content ? Why ?
Response: No data about DW nor water content were obtained
288-290 SAMPLING TIMING PROCESSING: explain clearly the sampling and extraction steps
This is confusing and it should be improved
Following paragraphs should be ordered according to the workflow: sampling, transfer to lab, extraction, assay …
Although apparently aminoacid composition and phenolics were assayed from the same subsample (see lines 288-290), different extraction methods are reported. see 327 and 374 .
Please explain ?
Response: Yes, it was a mistake. This information was clarified in the text. The amino acid analysis was started from frozen samples, while the phenol analysis was started from freeze-dried material. L311-314
301 - Missing biblio ref Malawi et al 2003
Response: Reference was added. L325
306 - correct: grinding
Response: Done. L330
306-307 5 ml final volume, P-buffer containing 100 mM KCl etc… Or were they added to the 5 mL ?
Response: The final volume was 5 ml per sample. This sentence was modified to improve clarity. L331
321 - merge the two short sentences
Response: Done. L346-347
327 - according to the method… modified as follows
Response: Done. L351
328 - v/v ? specify
Response: volume/volume. This abbreviation was specified the first is mentioned in the text. L298
332 - dry extract dissolved
Response: Done. L356
345 - no analysis only single parameter, therefore better just chl a fluorescence
Response: Done. L369
349 - OJIP was not introduced? explain or leave out.
Response: A graph showing OJIP phases was added. Figure 4. L179
350 - rephrase
Response: Done. L375-377
NOTE introduction needed about Chl fluorescence, basic intro, parameters
Response: Done. L167-177
352 - This is not correct. Energy fluxes relate to a specific set of jip-test parameters (energy cascade), while fluorescence is related to photochemistry rather than photosynthetic activity (as CO2 fixation). However, this manuscript only reports measured F0 and Fm parameters, with no OJIP kinetics and related JIP-test.
Response: Thank you for the comment. It is true. That sentence was deleted and OJIP kinetic graph was added. L179
Importantly: how many replicate measurements, on how many plants /leaves? Time of day ?
Response: This information was added. One measurement was made on nine fully developed leaves from nine different plants from each treatment (n=9). The analysis was performed at midday. L77-179
360 - Timing ? Quick freezing and processing ?
Response: Yes, this information was clarified in the text. L385-386
According to the stat anal paragraph, 3 determinations were made for each single parameter:
3 reps may not be enough, generally speaking. But does this mean 3 replicate assays on a single extract ? Or is it 3 different extracts?
Response: This information was clarified in the text. For all parameters analyzed, the mean and standard error were calculated from nine data (n=9). In the case of the biochemical measures, these nine data arose from combining 3 different sample extracts by 3 biochemical replicates. Each analytic was repeated in triplicate to confirm the results. L417-419
362 - missing biblio ref
Response: The reference was included. L386
362-364 - The cited method reference, points to another paper by Cubi 2008 and it is based on NBT.
Please check, explain and correct accordingly.
Morever this protocol starts from quick liquid N2 freezing, please describe the workflow clearly.
I cannot see where the mentioned reagents (hydroxylamine sulfanilic acid and α-1-naphthylamine) are used.
Check the text and the method.
Response: Thank you for the comment. The previous reference was a mistake. The reference was changed and the explanation of the methodology was improved. L385-386
CONCLUSION
402-403 Authors should only refer to the “T+H” treatment since no “T” alone treatment was included in the experimental design. In this case ”reduction” caused by H treatment or is it increase in the case of “T+H” ? Proper Control missing, therefore text should be modified accordingly.
Response: The conclusion was rewritten to make it more in accordance with the experimental design. L426-432
404-409 check language
Response: Done. L428-432
Round 2
Reviewer 1 Report
The authors made all required corrections.
Author Response
Thank you for your comments
Reviewer 2 Report
The manuscript has been certainly improved in this second version and some inaccuracies have been corrected.
However there are some critical points which in my opinion still need to be addressed properly.
For this reason, once again I would recommend a “major revision” needing accurate peer review.
About the english language: briefly, it is readable and understandable (mostly) but it still needs to be improved (see e.g. line 16 in the “Abstract”: to attempt its potential mode …; line 38 in the “Introduction”: ALS is key in the synthesis … and so on) and a number of faults need to be corrected.
Regarding the scientific content, my main concern lies in the chlorophyll fluorescence results, which I already pointed out in my comments to the first version of the manuscript.
In this second version the authors attempted to address my comments by adding a Fig. 4 with the OJIP kinetics. Oddly enough, the legend to Fig. 4 is wrong as it appears to be a duplicate of the legend to Fig 3 “ALS activity”.
Most importantly, though, the fluorescence kinetics plotted in this chart do not look right, and the O; J; I and P points do not seem to correspond to the inflection points of a typical fluorescence induction kinetic.
Therefore, I would ask the authors what are the data plotted in this chart ?
If the F0 and Fm values (needed for Fv/Fm calculation) where obtained from these fluorescence measurements I recommend to check the whole data set and possibly repeat the analysis of chl fluorescence data. The Fv/Fm values may need to be checked.
If no JIP-test parameters are presented, references to OJIP kinetics and to JIP-test may be omitted (see lines 167-177 and lines 374-375)
Balabanova et al. 2016 reported OJIP kinetics data for a very similar experiments on sunflower treated with Imazamox. Since I have no connection with the authors of this article, I feel free to point to the relevant bibliographic reference: ( Balabanova et al. (2016) Photosynthetic Performance of the Imidazolinone Resistant Sunflower Exposed to Single and Combined Treatment by the Herbicide Imazamox and an Amino Acid Extract. Front. Plant Sci. 7:1559. doi: 10.3389/fpls.2016.01559 ). The presentation of the OJIP kinetics and fluorescence data analysis may be useful as a guideline.
In any case, some theory behind chlorophyll fluorescence measurements should be given and explained in the “Introduction” section.
The following comments to the previous version of this manuscript, were not addressed:
“The chlorophyll fluorescence technique should be introduced more clearly in order to be useful to every reader. This paragraph must be greatly improved. Reference is made to the OJIP kinetics but no data are presented, therefore there is no need to mention the JIP test parameters.
About this point I have to ask a question: According to authors OJIP kinetics were recorded, therefore why no JIP-test data are presented?
These would add a lot more information and allow for a better understanding of the physiological effects of the treatments, thus substantially improving the manuscript.
I would recommend to include these data.”
line 130: ALS enzyme is the target of Imazamox action, therefore ALS activity is the main indicator of the effectiveness of herbicide treatment, rather than “one of the most reliable indicators”. rephrase
line 133: Following the above comment, in my view, ALS activity was depressed in herbicide treated plants rather than increased by the combined action of herbicide + biostimulant. One may suggest that the biostimulant protected the ALS activity (maybe keeping it close to pre-treatment level), rather than increased. In this case a non treated Control may explain the effect of the herbicide treatment.
Same comment applies to line 146 “resulted in an increase” and should be checked in the whole manuscripts to avoid misinterpretation. See also line 212,
Line 208 should point directly to cited biblio refs: rather than generic “herbicide detoxification”, define non-target Imazamox resistance.
Plant cultivation: the authors added details in lines 286-289 but I do not think that this explanation rules out the interference of salt stress to the plants. On the contrary, if the nutrient solution was used to irrigate plants every 3 days (see line 289), this would continuously supply more salts thus increasing salt accumulation into the potting substrate.
If this was not the case, authors should explain more clearly the plant cultivation conditions.
Pulsar 40 herbicide contains Imazamoz at 40 g/Liter (rather than 4% v/v see line 297).
The statistical analysis paragraph is still confusing and it should be improved/re-written more clearly. Actually, the authors replied to my previous comment but I do not understand their answer.
“ For all parameters analyzed, the mean and standard error were calculated from nine data (n=9). In the case of the biochemical measures, these nine data arose from combining 3 different sample extracts by 3 biochemical replicates. Each analytic was repeated in triplicate to confirm the results.”
Please explain that step by step. If determinations were done in triplicate, how could means be calculated from nine values ?
What is the meaning of “combining 3 different sample extracts by 3 biochemical replicates” ?
Author Response
Thank you for your constructive and perceptive comments, which have been useful in strengthening weak points and have greatly improved the quality and clarity of the work.
The following comments/actions (RESPONSE) are in response to your suggestions. The changes are highlighted in yellow in the manuscript
The manuscript has been certainly improved in this second version and some inaccuracies have been corrected.
However there are some critical points which in my opinion still need to be addressed properly.
For this reason, once again I would recommend a “major revision” needing accurate peer review.
About the english language: briefly, it is readable and understandable (mostly) but it still needs to be improved (see e.g. line 16 in the “Abstract”: to attempt its potential mode …; line 38 in the “Introduction”: ALS is key in the synthesis … and so on) and a number of faults need to be corrected.
Response: The English language was checked, and corrections were made in the indicated lines and in other parts of the manuscript
Regarding the scientific content, my main concern lies in the chlorophyll fluorescence results, which I already pointed out in my comments to the first version of the manuscript.
In this second version the authors attempted to address my comments by adding a Fig. 4 with the OJIP kinetics. Oddly enough, the legend to Fig. 4 is wrong as it appears to be a duplicate of the legend to Fig 3 “ALS activity”.
Most importantly, though, the fluorescence kinetics plotted in this chart do not look right, and the O; J; I and P points do not seem to correspond to the inflection points of a typical fluorescence induction kinetic.
Therefore, I would ask the authors what are the data plotted in this chart ?
Response: Thanks for the comment, the graph type previously used was not correct and therefore the data were not well represented. In the new version of the manuscript, the graph type has been changed and the figure legend has been corrected.
Lines 194-196
If the F0 and Fm values (needed for Fv/Fm calculation) where obtained from these fluorescence measurements I recommend to check the whole data set and possibly repeat the analysis of chl fluorescence data. The Fv/Fm values may need to be checked.
If no JIP-test parameters are presented, references to OJIP kinetics and to JIP-test may be omitted (see lines 167-177 and lines 374-375)
Balabanova et al. 2016 reported OJIP kinetics data for a very similar experiments on sunflower treated with Imazamox. Since I have no connection with the authors of this article, I feel free to point to the relevant bibliographic reference: ( Balabanova et al. (2016) Photosynthetic Performance of the Imidazolinone Resistant Sunflower Exposed to Single and Combined Treatment by the Herbicide Imazamox and an Amino Acid Extract. Front. Plant Sci. 7:1559. doi: 10.3389/fpls.2016.01559 ). The presentation of the OJIP kinetics and fluorescence data analysis may be useful as a guideline.
Response: Thanks for the reference. Fv/Fm values were deleted from table 2 and included in a new figure (Fig. 5). This figure includes the results for JIP-test parameters. The results and discussion have been rewritten according to the new figure. Lines 176-190; 204-208
In any case, some theory behind chlorophyll fluorescence measurements should be given and explained in the “Introduction” section.
Response: Done. Lines 48-54
The following comments to the previous version of this manuscript, were not addressed:
“The chlorophyll fluorescence technique should be introduced more clearly in order to be useful to every reader. This paragraph must be greatly improved. Reference is made to the OJIP kinetics but no data are presented, therefore there is no need to mention the JIP test parameters.
About this point I have to ask a question: According to authors OJIP kinetics were recorded, therefore why no JIP-test data are presented?
These would add a lot more information and allow for a better understanding of the physiological effects of the treatments, thus substantially improving the manuscript.
I would recommend to include these data.”
Response: JIP-test parameters were included in a new figure (Fig. 5) and the description of these parameters was included in the material methods section. Lines 203; 406-418
line 130: ALS enzyme is the target of Imazamox action, therefore ALS activity is the main indicator of the effectiveness of herbicide treatment, rather than “one of the most reliable indicators”. rephrase
Response: Done. Line 136
line 133: Following the above comment, in my view, ALS activity was depressed in herbicide treated plants rather than increased by the combined action of herbicide + biostimulant. One may suggest that the biostimulant protected the ALS activity (maybe keeping it close to pre-treatment level), rather than increased. In this case a non treated Control may explain the effect of the herbicide treatment.
Same comment applies to line 146 “resulted in an increase” and should be checked in the whole manuscripts to avoid misinterpretation. See also line 212,
Response: The text in the cited lines and in other parts of the discussion have been rewritten to avoid misinterpretation.
Line 208 should point directly to cited biblio refs: rather than generic “herbicide detoxification”, define non-target Imazamox resistance.
Response: Done. Line 234
Plant cultivation: the authors added details in lines 286-289 but I do not think that this explanation rules out the interference of salt stress to the plants. On the contrary, if the nutrient solution was used to irrigate plants every 3 days (see line 289), this would continuously supply more salts thus increasing salt accumulation into the potting substrate.
If this was not the case, authors should explain more clearly the plant cultivation conditions.
Response: There was no salt accumulation because every three days the nutrient solution in the trays was removed, the trays were thoroughly cleaned using distilled water, and new nutritive solution was added. This information was added to the text. Lines 340-342
Pulsar 40 herbicide contains Imazamoz at 40 g/Liter (rather than 4% v/v see line 297).
Response: Corrected. The correct form is weight/volume. Line 324
The statistical analysis paragraph is still confusing and it should be improved/re-written more clearly. Actually, the authors replied to my previous comment but I do not understand their answer.
“ For all parameters analyzed, the mean and standard error were calculated from nine data (n=9). In the case of the biochemical measures, these nine data arose from combining 3 different sample extracts by 3 biochemical replicates. Each analytic was repeated in triplicate to confirm the results.”
Please explain that step by step. If determinations were done in triplicate, how could means be calculated from nine values ?
What is the meaning of “combining 3 different sample extracts by 3 biochemical replicates” ?
Response: Yes, the above description could lead to confusion. This description has been modified in the new version of the manuscript. For each analysis, 3 subsamples were obtained from the frozen or lyophilized material. Then, from each of the sub-samples, the analysis was repeated in triplicate. Thus, 3 subsamples x 3 replications each = 9 data in total for each treatment. Lines 340-342 and 457
Round 3
Reviewer 2 Report
The authors significantly improved the manuscript in the present revision and faults/inaccuracies previously pointed out have been addressed.
I would recommend some minor revisions as follows:
Lines 46-54. This part could be improved and expanded. Do all classes of herbicides cause oxidative stress? Any reported effects of Imazamox based herbicides on ROS production/oxidative damage? Cite relevant biblio references.
Introduction to Chl fluorescence and OJIP analysis should be improved since not all readers are familiar with this technique.
Figure 4. The fast fluorescence kinetics are now correctly represented.
I would recommend to shift to the right the letters O; J; I,and P in the chart in at the corresponding time points on the x-axis (50 µs; 2 ms; 30 ms; 1 s respectively).
Since these fluorescence kinetics were apparently double O-P normalized, this should be declared both in the main text and in the legend to Fig 4, and the formula should be reported in Materials and Methods. Results should also be presented and discussed accordingly.
Figure 5. Authors should also explain how data presented in Fig. 5 (radar plot) were processed in order to be plotted. Values on the y-axis are …?
Author Response
The authors significantly improved the manuscript in the present revision and faults/inaccuracies previously pointed out have been addressed.
I would recommend some minor revisions as follows:
Lines 46-54. This part could be improved and expanded. Do all classes of herbicides cause oxidative stress? Any reported effects of Imazamox based herbicides on ROS production/oxidative damage? Cite relevant biblio references.
Response: This information was extended in the introduction. A wide range of herbicide types causes oxidative stress. However, no studies were found relating ROS generation and imazamox-based herbicides. (L47-52)
Introduction to Chl fluorescence and OJIP analysis should be improved since not all readers are familiar with this technique.
Response: More information about Chl fluorescence was added. (L53-63)
Figure 4. The fast fluorescence kinetics are now correctly represented.
I would recommend to shift to the right the letters O; J; I,and P in the chart in at the corresponding time points on the x-axis (50 µs; 2 ms; 30 ms; 1 s respectively).
Since these fluorescence kinetics were apparently double O-P normalized, this should be declared both in the main text and in the legend to Fig 4, and the formula should be reported in Materials and Methods. Results should also be presented and discussed accordingly.
Response: Changes in O, J, I, and P letters were performed as suggested. The data were not double O-P normalized. The graph represents the native fluorescence induction curve. This was clarified in the legend. (L208)
Figure 5. Authors should also explain how data presented in Fig. 5 (radar plot) were processed in order to be plotted. Values on the y-axis are …?
Response: The results of these parameters are shown in Figure 5. The data represented were obtained by dividing the mean obtained in treatment H + B by the mean obtained in treatment H + B to normalize the data and enable the comparison of parameters of different scales. The y-axis represents the number of times that the parameter is increased or decreased in the H + B treatment plants compared to the H treatment plants. For instance, 0.5 means that the parameter value was half as high for H + B plants as for H plants and 2 means that the value of this parameter was twice as high as in the H+B treatment. The description was improved in the figure and in material and methods. (L218-220; 428-431)